# Milk Production, N Partitioning, and Methane Emissions in Dairy Cows Grazing Mixed or Spatially Separated Simple and Diverse Pastures

**DOI:** 10.3390/ani10081301

**Published:** 2020-07-30

**Authors:** Lorena Carmona-Flores, Massimo Bionaz, Troy Downing, Muhammet Sahin, Long Cheng, Serkan Ates

**Affiliations:** 1Department of Animal and Rangeland Sciences, Oregon State University, Corvallis, OR 97333, USA; lorena.carmonaflores@oregonstate.edu (L.C.-F.); Massimo.Bionaz@oregonstate.edu (M.B.); troy.downing@oregonstate.edu (T.D.); 2Department of Crop and Soil Science, Oregon State University, Corvallis, OR 97333, USA; muhammet.sahin@oregonstate.edu; 3Faculty of Veterinary and Agricultural Sciences, Dookie Campus, The University of Melbourne, Dookie, VIC 3647, Australia; long.cheng@unimelb.edu.au

**Keywords:** herbal pasture, dairy cattle, greenhouse gas emissions, N partitioning, plant secondary compounds

## Abstract

**Simple Summary:**

It is possible to increase performance of grazing dairy cows while decreasing their environmental impact through incorporating high-quality pasture species that synthesize high bioactive compounds in diverse pastures. In this study, cows grazed combination of simple pastures containing perennial ryegrass and white clover or diverse pastures containing forbs known to contain bioactive compounds that can reduce methane emissions and nitrate leaching soils. The two types of pastures were grown either as mixed or spatially separated in adjacent strips. Our study indicated that spatial separation had little impact on cows’ performance, but pastures containing diverse plant species improved the production of protein and lactose in milk and reduced environmental impact, specifically by improving nitrogen utilization of cows and reducing the emission of methane per amount of feed eaten by the cows.

**Abstract:**

Increasing pasture diversity and spatially separated sowing arrangements can potentially increase the dry matter intake of high-quality forages leading to improved animal production. This study investigated the effects of simple (two-species) and diverse (six-species) pastures planted either in mixed or spatially separated adjacent pasture strips on performance, N partitioning, and methane emission of dairy cows. Thirty-six mid-lactation Jersey cows grazed either (1) simple mixed, (2) simple spatially separated, (3) diverse mixed, or (4) diverse spatially separated pastures planted in a complete randomized block design with three replicates. Compared to simple pasture, diverse pasture had lower CP content but higher condensed tannins and total phenolic compounds with an overall positive effect on yield of milk solids, nitrogen utilization, including a reduction of N output from urine, and methane yields per dry matter eaten. The spatial separation increased legume and CP content in simple pasture but decreased NDF in both diverse and simple pastures. In conclusion, increasing diversity using pasture species with higher nutritive value and secondary compounds can help improving the production while decreasing the environmental effect of dairy farming, while spatial separation had a minor effect on feed intake and yield, possibly due to overall high-quality pastures in early spring.

## 1. Introduction

Sustainable intensification in dairy farming is becoming increasingly important from economic viability, environmental health, and food security standpoints [1]. Providing high-quality pastures to grazing cows is pivotal to achieve high production efficiency in pasture-based dairy farming. Maintaining high forage production and quality in grass dominated pastures often requires intensive management and applications of P and N fertilizers at high rates. However, grazing systems that rely on high inputs can cause significant soil acidification, water pollution through NO_3_ leaching, and greenhouse gas (CO_2_, CH_4_, and N_2_O) emissions [2]. Furthermore, the N content of pastures often exceeds the requirements of grazing animals, leading to low N use efficiency [3,4]. Excess N intake by grazing cows predominantly (75–85%) is excreted in the form of urea in urine [5]. In addition to the metabolic cost of producing urea, the excretion of excess N in urine causes increased N_2_O emissions and NO_3_ leaching from pastures [6].

Diversification of pastures through inclusion of pasture herbs can reduce the urinary N excretion through lower dietary N intake and improved utilization of N in rumen [7]. It is possible to reduce the detrimental effects of intensive dairy farming through application of novel pasture systems and use of plants with high nutritive value and secondary metabolites [6,8,9]. Chicory with its pectin content may provide a highly effective synchronization of the fermentable carbohydrates and N in the rumen leading to improved rumen fermentation and lower nitrogen excretion from ruminants [10]. Similarly, grazing plantain causes less intensive N_2_O emissions from cows. This is attributed to its diuretic effect leading to diluted urinary N concentration than simple grass clover pastures [7,11]. Plantain contains bioactive compounds such as aucubin and catapol that reduce the urinary N concentration and increase the frequency of urination through altering kidney function [11]. 

Competition for resources and selective grazing of livestock usually reduce the proportion of the desirable pasture species over time [12]. A number of studies reported several agronomic and nutritional benefits of planting pure pastures of grass and clover in adjacent plots within the same field (spatial separation) as opposed to growing intermingled grass-clover mixtures [13,14]. Spatial separation of pasture species eliminates the interspecies competition for light, energy, and nutrients under selective grazing conditions and increases the proportion of more desired, but less competitive plant species [15]. Furthermore, the nutritional benefit of offering grasses and legumes in spatially separated adjacent plots can be similar to a pure legume monoculture [13]. 

Several grazing studies have indicated that spatial separation of pasture species substantially increases meat and milk yield [15,16]. The increased animal performance in spatially separated pastures has been attributed to improved diet quality and intake through minimized energy cost of foraging and selection associated with a mixed pasture sward. While these recent studies primarily investigated combinations of simple (two- or three-species) pastures, the combined effects of diversification and spatial separation on pasture and livestock production have not been evaluated. In particular, spatial separation can enable farmers to have better control over the botanical composition of specifically designed pasture mixtures (or “chemoscapes”) with high bioactive compounds [9,17]. 

The present study aimed to investigate feed intake, milk yield and composition, N partitioning, several blood parameters associated with glucose, protein, and lipid metabolism and liver and kidney status, and CH_4_ emissions from cows grazing simple and diverse pasture mixtures, grown together or in spatially separated adjacent plots. It was hypothesized that increased diversity (six-species pasture mix) and spatial planting arrangement of pasture species with high concentrations of bioactive compounds would increase feed intake and milk yield, improve the use of N reducing excretion, and reduce CH_4_ emissions of cows in pasture-based dairy farming. The overall objective was to develop sustainable, high-performing pasture-based dairy production systems where desirable pasture traits for animal performance are maintained at a high level in the diet.

## 2. Materials and Methods 

The study was conducted at the Oregon State University Dairy Center in Corvallis, Oregon, USA (44°34′N, 123°18′W 256 ft. a.sl.). All procedures were approved by the Institutional Animal Care and Use Committee (ACUP# 5028) prior to the commencement of the experiment. 

### 2.1. Pasture Establishment and Experimental Design 

A 7.2-ha paddock was divided into three 2.4-ha blocks to serve as replicates. Each block was divided into four 0.6 ha paddocks, which were randomly allocated to a combination of mixed or spatially separated simple and diverse pastures, giving a total of 12 grazing paddocks. Pasture treatments were: (1) A mixed perennial ryegrass cv. Kamo (*Lolium perenne*) + white clover cv. Seminole (*Trifolium repens*); (2) a spatially segregated perennial ryegrass + white clover; (3) a mixed diverse pasture consisting of festulolium cv. Perun (X *Festulolium braunii*), perennial ryegrass, white clover, chicory cv. Antler (*Cichorium intybus*), plantain cv. Boston (*Plantago lanceolata*), and birdsfoot trefoil cv. Bruce (*Lotus corniculatus*) or; (4) a spatially separated diverse pasture consisting of festulolium, perennial ryegrass, white clover, chicory, plantain, and birdsfoot trefoil (Figure 1). The four pasture combinations were sown on 20 October 2017, with 15-cm row spacing in a randomized complete block design with 3 replications with each block as a replicate (3 blocks for each treatment with 3 cows per block). In spatially separated, monoculture pasture, strips were sown in adjacent plots (0.1 ha) both in simple and diverse pastures.

### 2.2. Animals and Grazing Management

The grazing experiment was carried out from 3 April to 24 April 2019. Thirty-six Jersey multiparous (*n* = 27) and primiparous (*n* = 9) cows in mid-lactation were used in a randomized complete block design with 4 pasture treatments (9 cows per treatment) and three replicates (blocks). Nine cows were allocated to each treatment with 3 cows assigned to each replication. Cows were blocked for age (mean ± s.d.; 3 ± 1.3 years), live weight (mean ± s.d.; 492 ± 48.1 kg), and days in milk (mean ± s.d.; 169 ± 63.0 d). Each group of cows contained 1 primiparous and 2 multiparous cows. Dairy cows were offered a dietary treatment of: (1) A mixed, simple pasture; (2) a spatially separated, simple pasture; (3) a mixed, diverse pasture or; (4) a spatially separated, diverse pasture. Each group of 3 cows (2 multiparous and 1 primiparous cows) was randomly assigned to 1 of 12 0.6-ha pasture plots, where they rotationally grazed within the same plot at a stocking rate of 5 cows/ha. During the 21-d grazing period, the first 14 d were used to adjust the cows to the assigned dietary treatments (adaptation period), and the last 7 d were used for experimental measurements. Spatially separated adjacent monocultures in both simple and diverse pastures were grazed commonly by their respective groups of cows, as one pasture at the same time. Cows were strip grazed and allocated an estimated 16 kg of dry matter (DM)/cow per day with a post-grazing residual of 1300 kg of DM/ha. Water troughs were moved as needed to ensure ad libitum access to water. Five days prior to the start of the experiment, cows grazed on a sacrificial tall fescue paddock together as one large group.

The cows were milked twice daily (approximately 5 AM and 6 PM) and offered a new pasture allowance after each afternoon milking. All cows received 2 kg DM of rolled grain mix (corn and barley mix 50:50) and 91 g/d/cow mineral mix offered in two equal portions, immediately after each morning and afternoon milking throughout the grazing experiment (adaptation and measurement periods). The grain mix contained an average of 9% of crude protein (CP), 13.4 MJ ME/kg DM metabolizable energy (ME), 12.4% of neutral detergent fiber (NDF), and 2.3% of ash. Mineral mix consisted of 17–21% calcium, 7% phosphorus, 8% magnesium, 1.65% sulfur, 20–24 mg/kg selenium, and 100 IU/kg vitamin A. 

### 2.3. Apparent DM Intake and Pasture Nutritive Value

#### 2.3.1. DM Intake

Group herbage DM intake (DMI) was estimated by determining pre- and post-grazing pasture mass with a rising plate meter (PM; Jenquip, Feilding, New Zealand) by collecting 120 measurements in each daily allocation of pasture during the measurement period. A total of 20 rising PM readings were taken across the area in each of the spatially separated pasture strips, giving a total 120 readings across the entire plots. The PM was calibrated by regression against pasture mass at varying heights by collecting 24 quadrats (each 0.25 m^2^, 12 pre-grazing and 12 post-grazing quadrats) per pasture mixture and monoculture strips during the transition period. Quadrats were cut to 3 cm residual height with electric hand shears. Apparent group DM intake of cows were calculated from herbage disappearance between pre- and post-grazing herbage and area allocated. Calibration curves for each treatment were generated by fitting a single line through all the data. 

#### 2.3.2. Nutritive Values

A total of 50–75 pre-grazing pluck samples were collected randomly by hand across each pasture plot (with a “zigzag” pattern) at 2-day intervals during the experimental period. Samples were collected within each plot before cows were turned onto fresh pastures. Subsamples were sorted into botanical components and dried at 65 °C for 48 h. Botanical composition of samples was then calculated on a dry weight basis. 

A well-mixed bulk sample was ground in a Wiley mill with a 1-mm stainless-steel sieve (Thomas/Wiley, Swedesboro, NJ, USA) for chemical analyses. Samples were analyzed following the official methods of analysis of Association of Official Analytical Chemists [18] for DM (method 930.15), ash (method 942.05), and ether extract (method 920.39). The CP (6.25 ×N) concentration of all samples was determined by the Kjeldahl method using LECO FP828 (MI, USA). The NDF and acid detergent fiber (ADF) were assayed according to the methods described by Van Soest et al. [19] using an Ankom^200^ Fiber Analyzer (ANKOM Technology Corp., Macedon, NY, USA). The NDF was assayed with a heat-stable amylase and sodium sulphite, but both NDF and ADF were expressed inclusive of residual ash. Metabolizable energy was calculated using following equations [20]. ME (MJ/kg DM) = 0.16 × DOMD% [(0.95 × DMD%) − 0.9)]. Dry matter digestibility (DMD%) was calculated using the following formula DMD% = 88.9 − (0.779 × %ADF). Total apparent N intake was calculated using individual N contents (%) of the pasture on offer and concentrate × the average daily apparent DM intake (kg DM) of each component. Similarly, apparent metabolizable energy intake (MEI) was calculated by multiplying the total feed intake by the dietary metabolizable energy content (MJ/kg DM) of the pasture on offer and concentrate.

#### 2.3.3. Condensed Tannins and Total Phenolic Compounds

Condensed tannin (CT) were analyzed on samples of pastures based on the butanol-HCl-acetone method [21]. Tannin standards for the spectrophotometric assay were isolated from samples of tannin-containing plant material following a previously published protocol [22]. The scaled-down method previously described [23] was used to quantify total phenolic compounds (TP). Total phenolics were extracted from 50 mg of plant tissue with 1 mL of 50:50 (*v*/*v*) methanol:water on a rotator for 30 min followed by centrifugation at 16,000× *g* for 5 min at 4 °C. Then, 50 μL of supernatant from the sample extract were combined with 850 μL of SDS/TEA before adding 200 μL of FeCl3. Absorbance at 510 nm was read after a 15-min incubation period.

### 2.4. Foraging Behavior

Foraging behavior was scored by visual scanning of each cow at 15-min intervals from 8 AM to 4 PM on April 21 [24]. Each cow’s activity was scored for grazing, ruminating, and idling parameters by six observers. Positions of cows in spatially separated pastures were also recorded to determine the grazing time of individual forage strips. Grazing was defined as when a cow was actively eating with head down. Cows were recorded as idling if they had no specific jaw movements either standing or lying down. Grazing time on mixed pasture stands and each forage monoculture strip, ruminating and idling time were converted from the observation scores and multiplied by 15, assuming the same behavior over the previous 15 min. Forage selection ratios were calculated for each pasture monoculture strip [25]. The selection ratios compared the proportion of individual forage monoculture consumed by cows (forage disappearance) to the proportion of that forage monoculture available in pasture on offer. The following formula was used to calculate the selection ratios:

Selection ratio = [(disappearance (DM eaten) of a forage monoculture)/(disappearance (DM eaten) of all forage monocultures)]/[(pre-grazing pasture mass of a forage monoculture)/(average pre-grazing pasture mass of all forage monocultures)]. 

### 2.5. Milk and Body Condition Score Measurements

Daily milk yield and activity measurements were automatically recorded by an AfiMilk system (Kibbutz Afikim, Israel). Two milk subsamples were collected from each cow after AM and PM milkings on d 0 (baseline), 15, 18, and 21 to determine milk composition. Samples were analyzed commercially (Willamette DHIA Laboratory in Salem, OR) for fat, protein, lactose, solid non-fat (SNF) somatic cell counts (SCC), and milk urea nitrogen (MUN) by near-infrared spectrophotometry. Fat-corrected milk yield (FCM) was calculated using the following formula: 4% FCM = (0.4 × kg milk) + (15 × kg fat). Total daily milk N output was calculated by dividing the milk protein content (%) by 6.38. The N (%) was multiplied by the milk yield (kg/d) to obtain milk N output. Cows were scored weekly by two trained, independent evaluators using a five-point body condition score (BCS) scale (1 = thin; 5 = fat). 

### 2.6. Blood, Urine, and Fecal Measurements

#### 2.6.1. Sample Collection

Samples of blood, urine, and feces were collected immediately after the morning and afternoon milking on d 0 (baseline), 15, 18, and 21 of the experiment. Urine samples were collected midstream after manual stimulation of the vulva, acidified below a pH of 3.0 with sulfuric acid to prevent volatilization, and then stored at −20 °C until analysis. Feces samples were collected as cows defecated upon manual stimulation or by manual (grab) sampling and frozen at −20 °C until analysis. Blood samples (approximately 20 mL) were collected from the jugular vein. Samples were collected into evacuated tubes (Becton Dickinson Vacutainer Systems Becton Dickinson and Co., Franklin Lakes, NJ, USA) containing lithium heparin for plasma and without additives for serum. After blood collection, tubes with lithium heparin were placed on ice and tubes without additives were kept at room temperature to coagulate until centrifugation (~30 min). 

#### 2.6.2. Blood Analyses

Hematocrit was evaluated in samples collected in lithium heparin evacuated tubes. Serum and plasma were obtained by centrifugation at 1900× *g* for 15 min. Aliquots of serum and plasma were frozen (−20 °C) until further analysis. Plasma glucose, total cholesterol, non-esterified fatty acids, albumins, total bilirubin, aspartate aminotransferase, haptoglobin, creatinine, and urea were assessed using the ILab 600/650 (Instrumentation Laboratory, Bedford, MA, USA) by the Department of Animal Sciences Food and Nutrition (DIANA), Università Cattolica del Sacro Cuore (Piacenza, Italy) as previously described [26,27].

#### 2.6.3. Fecal and Urine N Analysis

##### Total N Content Analysis

Fecal samples were thawed, weighed, and dried in an oven at 55 °C for 72 h. Dry fecal samples were ground to 1 mm and analyzed for DM, ash, and N contents. The N contents of feces and urine samples were determined by using an N analyzer (LECO FP828, MI, USA). Ammonia (NH_3_) in plasma was determined using a commercial kit following the manufacturer’s instructions (MET-5086-CB, Cell Biolabs, San Diego, CA, USA). 

##### HPLC Analysis of Creatinine, Purine Derivates, and Urea in Urine

Subsamples of urine collected after the morning and afternoon milking from each cow on d 15, 17, and 21 were analyzed for concentration of creatinine, purine derivates (**PD**) allantoin and uric acid, and urea by using HPLC (Agilent 1260 Infinity, Agilent Technologies, Waldbronn, Germany) fitted with a Luna^®^ 5 µm C18(2) 100 Å, LC Column 250 × 4.6 mm (00G-4252-E0, Phenomenex, Torrance, CA, USA) and a SecurityGuard™ cartridges for C18 HPLC columns with 3.2 to 8.0 mm internal diameters (cat#AJ0-4287, Phenomenex). Urine samples were diluted 10-fold with double-distilled water and filtered using syringe filters and a 1 ml disposable luer lock syringe (57022-N04-C and 58901-S, MicroSolv Technology Corporation, Leland, NC, USA). Filtrated diluted samples were inserted into 1 mL transparent HPLC vials (82028-402, VWR, Radnor, PA, USA).

##### Urea in Urine

Urea was determined by fluorescence detection after derivatization using xanthydrol (cat#90-46-0, Alfa Aesar, Tewksbury, MA, USA) and following the gradient III and the automatic HPLC autosampler program of the method of Clark et al. [28] with modifications. Briefly, the run was 7 min with a full run (up to 12 min) every 10 runs using a blank to clean the column. The column was kept at room temperature (instead of 35 °C). The injection volume after derivatization was 8 µL (instead of 40.5 µL). Furthermore, despite that xanthydrol was solubilized in 1-isopropanol as indicated by Clark et al. [28], it separated quickly decreasing the derivatization of urea. To address the issue, we ran the second point of the standard curve every 10 runs, plus we used 3 samples that were added into the sequence every 10 samples and used the data to adjust for the final urea concentration. Quantitation of urea was achieved by a 5-points standard curve (4-fold dilution) of purified urea (BDH4602-500G, VWR) prepared in 2.4 pH double-distilled water to match the acidified urine.

##### Creatinine, Uric Acid, and Allantoin in Urine

Creatinine, uric acid, and allantoin concentration were analyzed using the same column as for the urea following the method described by George et al. [29]. A standard curve consisting of 480 µg/mL of allantoin (cat#97-59-6, Spectrum, New Brunswick, NJ, USA), 120 µg/mL of creatinine (cat#60-27-5, TCI, Portland, OR, USA), and 108 µg/mL of uric acid (cat#69-93-2, Alfa Aesar, Haverhill, MA, USA) diluted in 5-concentrations of 4-fold dilution was used for final quantification.

##### Urinary N Excretion and N Utilization for Milk Production

Urinary N excretion (g/d) was estimated as 21.9 (mg/kg) × BW (kg) × [1/urinary creatinine (mg/kg)] × urine N (g/kg), as previously described [30]. This formula assumes a constant production of creatinine. Since creatinine is secreted by the kidney at constant rate, the use of urinary creatinine help to account for the urine volume [30]. Due to high diurnal variation in urinary N concentration and excretion, a second urinary N estimation was also performed using the relationship with MUN as described by Nousiainen et al. [31]. The latter also accounts for the protein balance in the rumen and urinary-N excretion [31]. The milk samples were collected twice daily; thus, the concentration of MUN in milk allowed to account for the 24h period. The following formula was used to estimate urinary N output: 14.1 × MUN + 26. Milk N efficiency was calculated as N in milk divided by N intake. N efficiency for milk production was also calculated according to Nousiainen et al., [31] using the following formula: N-Efficiency, % = −0.73 × MUN + 38. The latter also accounts for the protein balance in the rumen and urinary-N excretion.

### 2.7. Methane Emission

Methane emission of individual cow was determined using the SF_6_ tracer method [32]. Due to insufficient number of canisters, CH_4_ emission was only measured in diverse mix, simple mix, and simple separated pasture treatments. Brass permeation tubes about 1 cm in diameter and about 4 cm long containing compressed SF_6_ gas were targeted to the rumen or reticulum and administered to cows with a bolus gun at the beginning of the trial. The release rate from the permeation tubes was about 1200 ng/min or 2 mg/d. Each permeation tube was loaded with 600 mg of SF_6._ The release rate of permeation tubes was measured gravimetrically for 6 weeks before placing them in the cow. 

A halter containing a collection system comprised of a filtered intake tube, capillary tubing, and an evacuated PVC collection canister was fitted to the animal, and the intake tube was positioned near the mouth and nose of the animal. The evacuated canister had a negative pressure (<0.2 psi), which drew air continuously for a 24-h period through the filter. After the sample was collected, the canister was removed and pressurized with high-purity nitrogen gas (N_2_). The collected gas was sampled and assayed using a gas chromatograph to determine the concentrations of CH_4_ and SF_6_. The emission rate of the permeation tube and the ratio of SF_6_ to CH_4_ in the collection canister were used to calculate the enteric emission rate of CH_4_ from the animal [32]. Samples were collected from six replications per treatment (only from the two multiparous cows in each grazing plot) for six consecutive days (on day 16 to 21). For the same six days, two ambient air samples (background concentrations) were collected in canisters located in a different paddock of the same pastures.

### 2.8. Statistical Analyses

All parameters except CH_4_ emissions were analyzed by analysis of variance (ANOVA) based on a 2 × 2 factorial model that accounted for the main effects of diversity and spatial separation in a complete randomized design. Treatment means for urine, feces, milk, and blood urea were determined using data collected from individual cows during the experimental periods (AM and PM of d 15, 18, and 21). CH_4_ emissions from cows were analyzed by ANOVA in a complete randomized design where three pasture types (simple mix, simple separated, and diverse mix) were the main effects. When available, data were corrected arithmetically by the baseline as previously described [33]. 

The average across the 3 cows in each plot was used as the experimental unit (pasture plot) rather than individual cows. Herbage intake and total DMI were estimated as means for the treatment group as cows grazed pastures in small groups.

The computations were carried out using GENSTAT statistical software version 18 (VSN International Ltd., Rothampstead, UK), with pasture type (simple vs. diverse) and spatial arrangement (mixed vs. separate) as fixed effects and block (*n* = 3/treatment) as random factor [34]. Significant differences among treatment means were compared by Fisher’s protected least significant difference at *p* ≤ 0.05. Tendency was declared with *p* ≤ 0.10.

## 3. Results

### 3.1. DMI and Grazing Behavior 

Averaged across the treatments, mean pre- and post-grazing pasture masses were 3166 and 1609 kg of DM/ha, respectively (Table 1). While there was no significant difference in pre- and post-grazing pasture masses between diverse and simple pastures, mixed pastures tended to have greater pre-and post-grazing pasture masses than spatially separated ones (*p* = 0.06; *p* = 0.09, respectively). The herbage DMI of cows ranged from 14.8 to 16.1 kg/cow/day, but neither pasture diversity (*p* = 0.73) nor spatial separation (*p* = 0.69) had a significant effect on DMI or on DMI corrected by BW. Similarly, cows had comparable metabolizable energy intake (MJ/day), regardless of the pasture type. Cows in diverse pastures had higher BCS compared to simple pasture. Cows grazing in spatially separated pastures had overall less activity compared to cows grazing in mixed pastures.

The overall means of grazing time, ruminating time, and idling time during the evaluation between morning and evening milking times were 228 min, 118 min, and 134 min, respectively (Table 2). Neither pasture diversity nor the spatial arrangement of the pastures had any significant effect on cow foraging behavior. However, the selection ratio indicated a preference for chicory and plantain over other forages. 

### 3.2. Pasture Nutritive Value and Botanical Composition

The diversity of pastures did not affect any nutritive value parameters except CP content of pastures (Table 3). Overall simple pastures had higher (*p* < 0.01) CP content than diverse pastures. There was also an interaction (*p* = 0.01) between diversity and spatial arrangement of the pastures for herbage CP content. There was no difference in CP among spatially separated and mixed in diverse pastures but spatially separated simple pastures had greater CP content than mixed simple pastures.

Overall, spatially separated pastures had lower (*p* = 0.05) NDF contents than mixed pastures. Similar to NDF content of pastures, spatially separated pastures tended to have lower (*p* = 0.06) ADF contents than mixed pastures. Increasing pasture diversity (*p* = 0.10) and spatial separation (*p* = 0.10) tended to result in lower EE contents. Neither diversity (*p* = 0.21) nor spatial separation (*p* = 0.89) had any effect on ash content of the herbage on offer. While diversity did not affect (*p* = 0.58) the ME content of pastures, spatial separation tended to increase (*p* = 0.06) the ME content of pastures. An interaction for TP were detected (*p* < 0.01). Total phenolic compounds were greater in spatially separated diverse pastures than all the other pasture combinations. Diverse pasture tended to have greater (*p* = 0.08) CT contents but spatial separation had no effect on CT content of pastures (*p* = 0.18).

Botanical composition of pasture on offer revealed significant differences (Table 4). Simple pastures had higher (*p* < 0.01) proportion of grass compared to diverse pastures (68.9% vs. 37.4%). However, total legume content was not affected by pasture diversity (*p* = 0.65). Average white clover content of pastures was higher (*p* < 0.05) in simple than diverse pasture. Spatial separation resulted in higher legume (*p* < 0.01) contents of pasture on offer. The increase in total legume content by spatial separation was almost 50%. Spatially separated pastures also tended (*p* = 0.07) to have greater white clover content than mixed pastures. While spatial separation did not affect the proportions of plantain and chicory, it tended to increase birdsfoot trefoil content (*p* = 0.08). 

The broadleaved weed content of pastures did not differ in relation to pasture diversity (*p* = 0.55) or spatial separation (*p* = 0.11). Although not measured, the predominant species of weed in all pasture was curly dock (*Rumex crispus*), in particular in spatially separated simple pastures. However, diverse pastures had higher (*p* < 0.05) dead material contents than simple pastures. Spatial separation tended to lead to lower dead material content (*p* = 0.09).

### 3.3. Milk Production and Composition

Average daily milk yield per cow was not affected by pasture diversity (*p* = 0.15) or spatial arrangement (*p* = 0.37) (Table 5). However, cows that grazed diverse pastures tended to have greater 4% FCM (kg/d) yield than those that grazed simple pastures (*p* = 0.08). Milk fat and protein contents and feed efficiency (FCM/DMI) were not significantly affected by pasture diversity or spatial separation (all *p* > 0.05). However, milk produced from diverse pastures tended (*p* = 0.08) to have higher SNF content than those from simple pastures. The cows that grazed diverse pastures produced 228 g/d more (*p* < 0.05) milk solids and 105 g/d more (*p* < 0.01) milk protein than those that grazed simple mixture pastures. Milk fat yield tended (*p* = 0.08) to increase in diverse vs. simple pasture and also tended (*p* = 0.10) to be greater in separated vs. mixed pastures.

Increased plant diversity in pastures led to larger (*p* < 0.05) lactose content in milk, while milk from spatially separated pastures had lower lactose content than mixed pastures (*p* < 0.01). Neither pasture diversity (*p* = 0.68) nor spatial separation (*p* = 0.42) had a significant effect on SCC. 

### 3.4. Measurements of N in Plasma, Urine, Feces, and Milk

#### 3.4.1. Dietary N Intake

A tendency for an interaction was detected between diversity and spatial arrangement of pastures for dietary N intake of cows (*p* = 0.06) (Table 6). Dietary N intake of cows on diverse pastures was similar regardless of the spatial arrangement (mixed or separated). However, cows that grazed spatially separated simple pastures consumed 122 g/d more N compared to those that grazed mixed simple pastures. 

#### 3.4.2. Urine N

Urine of the cows that grazed diverse pastures had lower N content and urea than the cows grazing simple pastures (Table 6). There was an interaction between pasture diversity and spatial separation for the urea concentration of urine (*p* < 0.01). Urine urea concentration in diverse pastures was similar regardless of the spatial arrangement but urine urea concentration of cows grazing spatially separated simple pastures was greater than mixed simple pastures. Concentration of creatinine and allantoin in urine tended to be lower in cows on diverse pastures than those that grazed simple pastures (*p* = 0.06 and *p* = 0.08, respectively). Consequently, the urine N output of the cows grazing diverse pastures was 53 g/d lower than those that grazed simple pastures.

Urine NH_3_ and creatinine concentrations of cows that grazed spatially separated pastures tended to be greater than of those grazed the mixed pastures (*p* = 0.05). Among PD, concentration of uric acid was greater (*p* = 0.05) while concentration of total PD (i.e., uric acid + allantoin) tended to be greater (*p* = 0.09) in urine of cows grazing spatially separated than mixed pastures. However, when level of PD was adjusted by the creatinine, no differences were observed between pastures. The total N excretion through urine was not affected by the spatial arrangements of pastures. 

#### 3.4.3. N in Feces

Feces N content (%) of cows that grazed diverse pastures tended to be greater than those grazed simple pastures (*p* = 0.09; Table 6). In addition, cows on spatially separated pastures had lower N content in their feces than those that grazed mixed pastures (*p* < 0.05).

#### 3.4.4. Milk N and Efficiency of Using N for Milk Production

Milk urea nitrogen content from diverse pastures was 3.6 mg/dl less compared to milk from simple pastures (*p* = 0.01; Table 6). A tendency for an interaction (*p* = 0.06) between pasture diversity and spatial separation was detected for MUN content, as spatial separation had opposite effects on MUN when adopted in diverse (where separation numerically decreased MUN) or in simple pasture (where separation tended to increase MUN). Milk N output was greater with cows that grazed diverse pastures as compared to those that grazed simple pastures (*p* < 0.05). However, the excretion of N through milk was not significantly affected by the spatial arrangement of pastures (*p* = 0.42). The cows grazed on diverse pastures had higher N use efficiency for milk production compared to simple pastures (*p* < 0.01 and *p* < 0.05). While spatial separation tended to decrease (*p* < 0.06 and *p* = 0.09) the N use efficiency in simple pastures, no difference was observed in diverse pastures. 

### 3.5. Blood Parameters

Most blood metabolites of dairy cows that grazed mixed or spatially separated simple and diverse pastures were similar (Table 7). The hematocrit decreased in cows grazed in spatially separated strips compared to mixed plants in simple pastures only. Besides hematocrit, blood urea and haptoglobin were also affected. Blood urea was lower in cow grazed on diverse vs. simple pastures. An interaction was detected between pasture diversity and spatial separation for urea (*p* < 0.01). Plasma urea concentration of the cows grazing diverse pastures was similar in spatially separated and mixed pastures, while cows that grazed spatially separated simple pastures had substantially higher blood urea than those that grazed mixed simple pastures (*p* < 0.01). Haptoglobin concentration in plasma was lower in cows grazing spatially separated pastures than those that grazed mixed pastures (*p* < 0.05). A tendency for an interaction was detected for albumin (*p* < 0.06) due to a different effect of strip pastures when cows were grazing on diverse or simple pasture. Cows grazing on strips had a numerically lower albumin when the pasture was diverse but larger albumin when the pasture was simple. 

### 3.6. Methane Emissions

The daily CH_4_ production per cow was 335 g/d, 393 g/d, and 382 g/d for diverse mix, simple mix, and spatially separated simple pastures, respectively (Table 8). There was no statistical difference for CH_4_ emissions of the cows in relation to production of milk and its components (all *p* > 0.05). However, cows that grazed mixed diverse pastures had less (*p* = 0.05) CH_4_ production per kg of DMI than those grazed mixed and spatially separated simple pastures.

## 4. Discussion

### 4.1. Increased Diversity in Pastures, but not Spatial Separation, Improves Milk Quality

#### 4.1.1. Possible Causes of the Positive Effect of Pasture Diversity on Milk Solids 

The findings of the current study indicated that increased diversity, particularly through the inclusion of pasture forbs, improved or tended to improve overall production of milk solids. Milk protein and lactose were increased, with a tendency for higher milk fat yield and overall improved FCM despites having lower CP in the diet. Although, in our study, we did not observe difference in milk yield, the tendency for a higher FCM is in line with the benefit of diverse pastures on milk yield in prior studies [7,13,36,37]. Superior milk production in the prior studies can be attributed to improved pastures quality by inclusion of forb and legume species that have lower NDF and higher NFC content as compared to grass species [38]. In our study, the diverse pastures were not different in NDF or NFC compared to simple pastures; this may partly explain the lack of significant effect on milk yield detected. Another reason for the lack of effect on milk yield by more diverse pastures in our study can be the overall high quality of our pastures. In previous studies, it was observed that the superiority of diverse pastures in supporting high animal performance was only seasonally effective [39,40] and the effect was altered by the daily forage allocation and concentrate supplementation [40,41]. Soder et al. [40] did not report an increase in milk yield from cows that grazed diverse pastures as compared to those grazed simple pasture possibly due to the high levels of concentrate (40%) supplemented in the diet. Similarly, Woodward et al. [39] reported that the increase in milk production from diverse pastures was only apparent during fall grazing period due to an increase in the chicory content of pastures. 

The larger yield of milk solids in cows grazed on diverse vs. simple pastures in our study can be partly explained by the level of CT and a possible better rumen situation by the presence of chicory and/or plantain in the diverse pastures, as supported by the larger milk protein yield (see Section 4.2 and Section 4.3 below). It has been reported that CT present in birdsfoot trefoil can increase milk protein [42], as they bind to the proteins in the plant, and inhibit the activity of microorganisms. This reduces protein degradation in rumen, increasing the absorption of amino acids in the small intestine. In agreement with this, we detected a tendency for higher CT in diverse vs. simple pastures. We were expecting none or very little presence of CT in simple pastures. However, the detected CT levels in simple pastures in our study is likely due to the high proportion in the broadleaved weed of curly dock, known to have large amount of CT [43], which was however not grazed by the cows. Thus, our data likely underestimated the difference in CT between pastures. The higher milk solids, including protein, can be also due to the presence of chicory and plantain in the diverse vs. simple pastures. Results from our study agrees with a prior study reporting higher milk solids yield due to an increase in the milk protein content of cows that grazed pasture containing chicory and plantain vs. pasture with only ryegrass and white clover [37]. Similar to our study, in that prior study, the use of forb, particularly chicory, increased milk lactose; however, the reason for such an effect is unclear. 

In their review papers, Pembleton et al. [41] stated that the positive response of cows to increased availability of high-quality forages in diverse pastures may not be obvious at high-forage allocations due to selection opportunity of the grazing cows [36,40]. The lack of difference in DMI between simple and diverse pasture can also be partly explained by the low DM content in the herbs in the diverse pasture [37] despite the fact that the selection index indicated that cows had a strong preference for chicory and plantain. Similarly, Gregorini et al. [44] reported that cows that were offered chicory and plantain exhibited less rumination time and an increased idling time compared to perennial ryegrass. We did not observe any difference in grazing behaviors in our study. The lack of difference in the current study may be related to the temporal grazing pattern of dairy cows, as they may have consumed more preferred forages at night after they were offered fresh pastures while the grazing behavior measurement were conducted during daytime. 

#### 4.1.2. High Quality of Pastures Likely Masked Any Beneficial Effects of Spatial Separation on Milk Yield 

The basic argument of the value of spatial separation compared to growing intermingled grass-clover mixtures is increased intake of high-quality forages through minimized energy cost of foraging and selection associated with a mixed pastures [16]. Chapman et al. [13] reported that offering legumes and grasses in a 50:50 area ratio as free choice improved the feeding value of the pastures to a level at par with a pure legume monoculture. In the present study, we detected a higher quality of pastures driven by larger CP and lower NDF in spatially separated vs. mixed pastures. In spatially separated pastures the legume content of the herbage on offer was 16.5% greater than in mixed pastures. Despite the high quality of the herbage, spatial separation of pasture species in adjacent strips did not result in any differences in apparent feed intake and only marginally increased the milk fat yield. The lack of difference may be attributed to overall high nutritive quality of pastures regardless of the diversity and sowing arrangements indicating that the chemical and physical composition of pastures were already conducive to maximum feed intake. In support of this, Pembleton et al. [15] reported that spatially separated strips and intermingled mixtures of perennial ryegrass, white clover, and plantain pastures had greater milk solid production than perennial ryegrass monoculture and this increased milk production was attributed to increased intake of higher nutritive value pastures, but this effect varied across seasons depending the pasture quality and the physiological conditions of the cows. 

### 4.2. Diverse Pastures Improve N Utilization

Excretion of N through urine, feces, and milk has a positive linear relationship with N intake [45]. Overall, urinary N increases at high levels of N intake particularly if the proportion of rumen-degradable protein is high and energy level in diet is not matched with N intake. In that condition, the deamination of amino acids by rumen bacteria creates a surplus of NH_3_, which is not utilized by the microbes, and therefore NH_3_ is detoxified by the liver in the form of urea that is excreted in urine [46].

It is important to note that caution is needed to interpret the estimated excretion of N in urine from spot urine samples, as performed in the present study, as diurnal variations in urinary N concentration and excretion are known to occur [47,48]. However, it is impractical to perform multiple diurnal urinary samples in a pasture-based system, except when automatic sample collection systems are available. For this reason, we have also estimated the N urinary excretion using the MUN. Despite this known limitation, our data clearly highlighted an important and consistent reduction of N excretion by diverse pastures containing chicory and plantain compared to simple pastures. In the present study, the lower excretion of N in urine, as well as the lower levels of urea in plasma, urine, and milk, in cows that grazed diverse vs. simple pastures can be explained by the lower CP levels in the former vs. the latter and the presence of forbs. 

The level of CP in the diet had likely a lower role in the observed effect. Cows that grazed diverse pastures had 40% lower urinary N output compared to those that grazed simple pasture mixtures, while the difference in CP was only approx. 14% between pastures. 

A more important role in the decrease of N output in diverse vs. simple pastures might have been played by the presence of chicory and plantain. Totty et al. [7] reported that diverse pasture containing chicory and plantain decreased urinary N by 20%. Mangwe et al. [49] reported lower urine urea concentrations from cows that grazed plantain (27.2 mmol/L) and chicory (29.2 mmol/L) compared to cows grazed on perennial ryegrass-white clover pastures (128.7 mmol/L), indicating a reduction of >70% urea in urine. In our case, the urinary N concentration decreased >30% with diverse vs. simple pastures. Decreased urinary N output and increased milk N output in cows grazed on diverse pastures with low CP suggest a shift in the utilization of N [7]. The presence of substantial amount of bioactive compounds, including diuretic molecules, is the main reason chicory and plantain reduce urea level and overall N output and partition the N toward milk in dairy cows [7,50]. Higher utilization of N accompanied by a decrease in urinary N output was observed in dairy cows when chicory was incorporated into pasture mixtures [7,51]. 

#### 4.2.1. Shift of N in Milk in Diverse Pasture is Partly Driven by Level of CT

The observed shift in N output from urine to milk and, with a tendency, to feces for the cows that grazed diverse compared with simple pastures can partly be explained by the tendency for higher CT on the former vs. the latter. Condensed tannins are known to decrease the degradability of protein in the rumen and, thus, they decrease deamination of amino acids resulting in lower NH_3_, as observed when CT-rich birdsfoot trefoil hay was fed to dairy cows [6,50,52]. Reduction of protein degradation by CT increases the level of rumen undegradable protein, which is generally associated with higher milk yield or FCM, although this effect on milk production is not consistent [53]. In grazing cows, the improvement of RUP is especially important [54]. Concurrent to the effect of CT, chicory and plantain are known to have high NFC content [55,56]. The NFC are readily available for bacteria to couple with the use of urea, increasing the efficiency of N utilization in the rumen. The energy-protein coupling (or nutritional synchrony) in the rumen plays a major role in the efficiency of the fermentation [57]. Although not statistically significant, the numerically greater CT and NFC in diverse vs. simple pastures might have helped to improve the N utilization in cows but cannot be the only explanation.

A better nutritional synchrony in the rumen would likely translate to higher microbial activity. This can be indirectly estimated by measuring urinary PD [55,58]. In the current study, we observed minimal effects of pasture types on PD in urine. The urine of cows that grazed spatially separated pastures containing forages with lower NDF contents (but similar NFC as compared to mixed pastures) had higher uric acid and tended to have higher total PD content, while increased diversity of the pasture tended to decrease allantoin and numerically decreased PD content in urine. The PD in urine is associated with the level of N intake in cows [59]. In our study, the PD level in urine had a similar trend as the N intake; however, despite a tendency for lower N intake in the cows in the diverse vs. simple pastures, the concentration in urine of PD:creatinine was not different between pastures. Creatinine is produced by muscles and, if the kidneys function normally and the animal is not sick, it is secreted into urine at a constant rate [60,61]; thus, it can be used to estimate daily urine volume [35]. For this reason, the correction of PD by creatinine provides a better estimate of the PD production. The lack of effect on PD:creatinine despite a tendency for lower intake of N might support a higher capacity of the microbes to use dietary nitrogen in diverse vs. simple pastures. The lack of effect on NH_3_ concentration or NH_3_:creatinine in urine does not seem to support a greater rumen microbial activity, since a higher efficiency of using N in rumen should results in lower NH_3_ in urine. However, it has been recently demonstrated that the use of urinary creatinine is not adequate to estimate urine volume output in dairy cows using spot sampling [62]. As indicated above, spot urine sampling is not ideal. Thus, a clear conclusion about rumen microbial activity using PD and other N parameters in urine in our study cannot be made. 

Based on the above, it appears that in our study the higher milk solid production in cows grazed on diverse vs. simple pastures was partly driven by the increased rumen undegradable protein due to the activity of CT on rumen protein. The data, although with the limitations highlighted above, do not support an effect of increased rumen efficiency on greater yields of milk solids in diverse vs. simple pastures. 

#### 4.2.2. Presence of Plantain in Diverse Pasture Might Improve Environmental Impact

The lower N in urine in cows grazed on diverse vs. simple pastures can be partly explained by the presence of plantain in the diverse pasture. Plantain can decrease the concentration of N in urine due to its diuretic effect [6,49]. A larger diuretic effect is supported by the tendency for a lower creatinine concentration in urine of diverse vs. simple pastures. The shift of N toward feces in diverse vs. simple pastures concurrent with a reduction of the amount of N in urine, which can be readily converted to NH_3_, has important environmental implications. Nitrogen in feces is in a less readily volatilized form than N in urine reducing N leaching; thus, diverse pasture can help to reduce N pollution [63].

### 4.3. Diverse Pastures Might Improve Nutrient Utilization

The metabolic parameters measured in blood were all in the expected ranges for the stage of lactation [64,65] and were not different between groups, including NEFA, suggesting a minimal or no effect of the treatments on the metabolism. Also, liver activity was normal in all cows and was not affected by type of pasture, considering parameters such as albumin, bilirubin, and AST/GOT, which were all in the expected ranges indicating an overall healthy liver [26,65,66]. However, the cows grazing diverse pastures had higher BCS compared to those grazed on simple pastures, indicating a better utilization of the available nutrients in the former vs. the latter. However, the lack of BCS measurements at baseline did not allow for determining change in BCS, which would be more accurate as measurement of change in adiposity. Cows used in the present experiment were in mid-lactation, where the adipose tissue becomes more anabolic and triglycerides starts to accumulate [67,68]. Thus, higher capacity of using nutrients would increase fat accumulation in the adipose tissue. The simultaneous presence of a similar DMI between cows in the various pastures, but numerically larger FCM and larger BCS in diverse vs. simple pastures supports a better utilization of the nutrient in the diet in the diverse vs. simple pastures. 

Among measured parameters in blood, haptoglobin was larger in cows grazed on mixed vs. spatially separated pastures. Haptoglobin is an acute phase protein and it is used as index of inflammation [26,69]. There are no reported ranges for this parameters for Jersey cows in mid-lactation; however, based on prior data [26,66,70] none of the groups could be considered to experience an inflammatory status. The reason for the observed effect is unclear.

### 4.4. Diverse Pastures Decrease CH_4_ Emissions

Similar to the findings on N partitioning, some positive effects of diverse pastures were also observed on CH_4_ production. Although the daily total CH_4_ production per cow was not altered by the pasture type, cows that grazed diverse pastures containing birdsfoot trefoil, chicory, and plantain had 20% lower CH_4_ production per kg of apparent DMI compared with cows that grazed simple pastures. These findings are in agreement with the results reported by Woodward et al. [71] where Holstein-Friesian cows that grazed birdsfoot trefoil had 17% lower CH_4_ emission per unit of DMI compared with cows that grazed perennial ryegrass. The decrease in CH_4_ production can be attributed to the presence of secondary metabolites that chicory and birdsfoot trefoil contain, especially CT [71,72]. 

## 5. Conclusions

The current study confirmed that increasing pasture diversity and spatial separation of pasture species in adjacent plots increase the proportion of desirable pasture species in the pasture on offer. In particular, the value of spatial separation was highlighted in diverse pastures where proportion of birdsfoot trefoil in botanical composition were substantially higher than diverse mixed pastures. The spatial sowing arrangements provide better establishment opportunities for plant species with novel nutritional characteristics, but weak agronomic attributes such as slow establishment, poor competition, or short life cycle, into livestock grazing systems. Increasing diversity provided greater milk solid and milk protein yields. However, animal DMI and milk yield responses to spatial separation of pasture species were not reflected in this study possibly due to overall high quality of pastures in early spring. It is likely that the benefit of spatial separation may be more obvious particularly when grass growth and quality decrease in late spring-summer period. The value of incorporating pasture forbs with high bioactive compounds was highlighted in increased milk solid production and reduced urinary N outputs and CH_4_ emissions per DM eaten by cows grazing diverse pastures. This indicates the potential benefit of creating “chemoscapes” in grazing fields or specifically designing pasture mixtures with high bioactive compounds in spatially separated plots for improving the animal yield and health of the environment.

## Figures and Tables

**Figure 1 animals-10-01301-f001:**
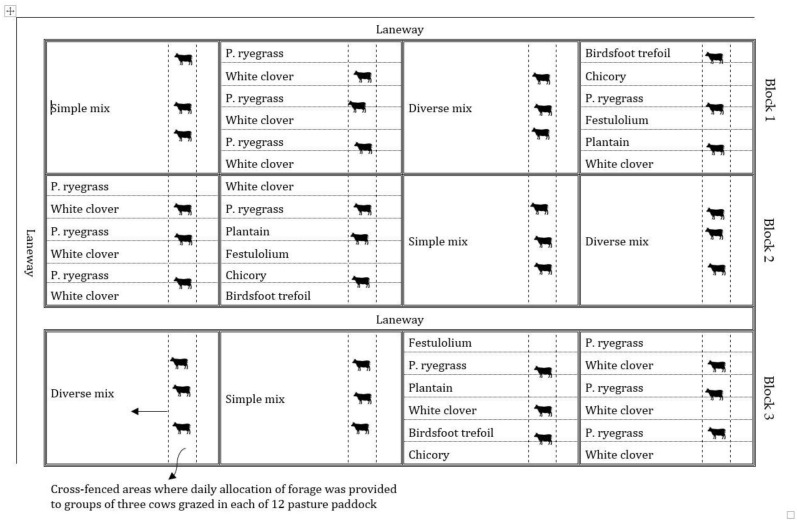
Diverse and simple pasture mixtures sown either as mixed or spatially separated plots.

**Table 1 animals-10-01301-t001:** Effect of pasture diversity (diverse or simple) and spatial arrangement (mix or separated) on herbage mass, apparent intake, and body condition.

Items	Diverse		Simple		SEM ^6^	*p* Values ^7^		
	Mix	Separated	Mix	Separated		D	SA	D × SA
Pre-GPM ^1^, kg of DM/ha	3321	3151	3237	2955	96.3	0.19	0.06	0.58
Post-GPM, kg of DM/ha	1729	1584	1764	1357	140	0.51	0.09	0.38
Herbage DMI ^2^, kg/cow/day	16.0	15.8	14.8	16.1	1.24	0.73	0.69	0.56
Total DMI, kg/cow/day	18.0	17.8	16.8	18.1	1.24	0.73	0.69	0.56
MEI ^3^, MJ/day	202	201	185	206	13.5	0.68	0.50	0.46
Herbage DMI/kg of BW ^4^, g/kg	3.3	3.2	3.1	3.2	0.28	0.63	0.96	0.67
Total DMI/kg of BW, g/kg	3.7	3.6	3.5	3.6	0.28	0.62	0.92	0.68
BCS ^5^	3.0	3.1	2.9	2.8	0.06	0.03	0.80	0.49
Activity, steps/h	199.4	181.7	212.2	186.3	5.93	0.15	0.01	0.50

^1^ Grazing pasture mass; ^2^ Dry matter intake; ^3^ Metabolizable energy intake; ^4^ Body weight; ^5^ Body condition score; ^6^ SEM: Standard error for interaction; ^7^ D: Diversity; SA: Spatial arrangement; D × SA: Interaction.

**Table 2 animals-10-01301-t002:** Effect of pasture diversity (diverse or simple) and spatial arrangement (mix or separated) on foraging behavior of cows.

Items	Diverse			Simple		SEM ^1^	*p* Values ^2^		
	Mix	Separated		Mix	Separated		D	SA	D × SA
**Foraging behavior, min**									
Grazing time	210	233		245	223	13.8	0.37	0.95	0.11
Ruminating time	125	110		118	120	12.1	0.89	0.58	0.49
Idling time	145	137		117	137	14.1	0.32	0.68	0.32
**Grazing time of forage strips, min**									
Total grass		95			108				
Total legume		58			115				
Perennial ryegrass		53			108				
White clover		18			115				
Birdsfoot trefoil		40							
Festulolium		42							
Chicory		55							
Plantain		23							
**Selection ratio (Post-grazing pasture mass, kg of DM/ha)**									
Perennial ryegrass		0.97 (1659)			1.10 (1398)				
White clover		0.98 (1414)			0.98 (1271)				
Birdsfoot trefoil		0.89 (1443)							
Festulolium		0.78 (2486)							
Chicory		1.20 (1208)							
Plantain		1.20 (1496)							

^1^ SEM: Standard error for interaction; ^2^ D: Diversity; SA: Spatial arrangement; D × SA: Interaction.

**Table 3 animals-10-01301-t003:** Effect of pasture diversity (diverse or simple) and spatial arrangement (mix or separated) on nutritive value of pastures.

Items	Diverse		Simple		SEM ^1^	*p* Values ^2^		
	Mix	Separated	Mix	Separated		D	SA	D × SA
Ash, % DM	9.0	10.2	9.2	8.1	0.70	0.21	0.89	0.15
CP, % DM	17.5 ^bc^	16.9 ^c^	18.3 ^b^	21.6 ^a^	0.27	0.01	0.01	0.01
ADF, % DM	20.6	19.7	22.5	19.0	0.99	0.58	0.06	0.24
NDF, % DM	35.9	32.9	41.8	33.4	2.39	0.22	0.05	0.30
NFC, % DM	34.9	37.6	27.8	34.1	2.83	0.11	0.16	0.54
EE, % DM	2.7	2.4	3.0	2.7	0.14	0.10	0.10	0.86
ME, MJ/kg DM	10.9	11.0	10.7	11.1	0.12	0.58	0.06	0.24
TP, mg/g	25.2 ^b^	83.9 ^a^	28.8 ^b^	32.7 ^b^	6.5	0.02	0.01	0.01
CT, mg/g	8.0	8.6	6.9	7.7	0.47	0.08	0.18	0.79

^a–c^ Means within a row with different superscripts differ (*p* < 0.05). ^1^ CP: Crude protein; ADF: Acid detergent fiber; NDF: Neutral detergent fiber; NFC: Non-fibrous carbohydrates; EE: Ether extract; ME: Metabolizable energy; TP: Total phenolic; CT: Condensed tannins; ^1^ SEM: Standard error for interaction; ^2^ D: Diversity; SA: Spatial arrangement; D × SA: Interaction.

**Table 4 animals-10-01301-t004:** Effect of pasture diversity (diverse or simple) and spatial arrangement (mix or separated) on botanical composition (% of total dry matter (DM)) of pastures.

Items	Diverse		Simple		SEM ^1^	*p* Values ^2^		
	Mix	Separated	Mix	Separated		D	SA	D × SA
Perennial ryegrass	10.3	12.5	75.9	61.9	5.9	0.01	0.35	0.22
White clover	14.5	18.1	20.0	34.1	4.1	0.04	0.07	0.24
Festulolium	34.3	17.8	-	-	2.8	-	0.05	-
Chicory	14.4	17.9	-	-	1.0	-	0.14	-
Plantain	20.7	11.0	-	-	4.2	-	0.24	-
Birdsfoot trefoil	1.0	16.3	-	-	3.4	-	0.08	-
Weeds	0.4	3.3	1.6	4.0	1.4	0.55	0.11	0.87
Dead material	4.4	3.0	2.6	0.0	1.0	0.05	0.09	0.54
Total legume	15.5	34.4	20.0	34.1	4.5	0.65	0.01	0.61

^1^ SEM: Standard error for interaction; ^2^ D: Diversity; SA: Spatial arrangement; D × SA: Interaction.

**Table 5 animals-10-01301-t005:** Effect of pasture diversity (diverse or simple) and spatial arrangement (mix or separated) on milk yield and composition.

Items	Diverse		Simple		SEM ^1^	*p* Values ^2^		
	Mix	Separated	Mix	Separated		D	SA	D × SA
**Yield ^3^**								
Milk, kg/d	23.8	25.7	22.7	23.0	1.16	0.15	0.37	0.49
4% FCM ^3^, kg/d	27.9	31.5	25.4	27.4	1.60	0.08	0.13	0.64
Milk solids ^4^, g/d	2234	2386	2095	2070	91.5	0.04	0.51	0.37
Milk fat, g/d	1228	1413	1090	1215	81.3	0.08	0.10	0.72
Milk protein, g/d	850	915	788	767	29.4	0.01	0.47	0.19
Feed efficiency ^5^	1.60	1.78	1.51	1.53	0.16	0.33	0.56	0.62
**Component**								
Fat, %	5.2	5.5	4.8	5.3	0.23	0.24	0.15	0.64
Protein, %	3.6	3.6	3.5	3.4	0.09	0.16	0.53	0.84
SNF ^6^, %	9.5	9.3	9.2	9.1	0.10	0.08	0.16	0.96
Lactose, %	4.8	4.7	4.7	4.6	0.03	0.03	0.01	0.23
SCC ^7^, log_2_	5.4	5.6	5.1	5.5	0.35	0.68	0.42	0.82

^1^ SEM: Standard error for interaction; ^2^ D: Diversity; SA: Spatial arrangement; D × SA: Interaction; ^3^ FCM: 4% fat-corrected milk, ^4^ g of solids non fat; ^5^ Feed efficiency (FCM/DMI), ^6^ SNF: Solids non fat, ^7^ SCC: somatic cell count log_2_ of 10^3^ cells/ml.

**Table 6 animals-10-01301-t006:** Effect of pasture diversity (diverse or simple) and spatial arrangement (mix or separated) on nitrogen partitioning in dairy cows.

Items ^1^	Diverse		Simple		SEM ^2^	*p* Values ^3^		
	Mix	Separated	Mix	Separated		D	SA	D × SA
N intake, g N/d	472	452	459	581	31.5	0.11	0.15	0.06
**Urine**								
N, %	0.30	0.29	0.45	0.50	0.022	0.01	0.41	0.29
NH_3_, mmol/L	2.74	3.21	2.70	3.57	0.39	0.59	0.05	0.50
Urea, mmol/L	104.0 ^c^	98.1 ^c^	132.0 ^b^	175.7 ^a^	5.77	0.01	0.17	0.01
Creatinine, mmol/L	2.35	2.81	2.69	3.21	0.15	0.06	0.02	0.86
NH_3_: Creatinine	1.21	1.21	1.16	1.18	0.09	0.69	0.93	0.95
Allantoin, mM	13.3	15.6	15.9	17.1	0.98	0.08	0.13	0.56
Uric acid, mM	0.60	0.84	0.74	0.82	0.05	0.33	0.05	0.21
Total PD, mM	13.8	16.7	16.7	17.8	1.05	0.11	0.09	0.43
Allantoin:creatinine	5.82	5.81	6.07	5.71	0.23	0.76	0.44	0.48
Total PD:creatinine	6.11	6.10	6.35	6.01	0.24	0.76	0.50	0.53
N output ^4^, g/d	166.9	147.1	195.0	220.4	9.80	0.01	0.78	0.06
N output ^5^, g/d	139.7	113.6	182.2	176.9	10.16	0.01	0.17	0.34
**Feces**								
N, %	3.7	3.6	3.7	3.5	0.04	0.09	0.02	0.62
Ash, %	19.8	21.5	19.5	21.1	0.77	0.64	0.04	0.93
DM, %	12.3	11.4	11.0	11.6	0.49	0.33	0.73	0.20
**Milk**								
MUN, mg/dl	10.0	8.6	12.0	13.8	0.69	0.01	0.78	0.06
N output, g/d	134.8	144.2	124.7	123.4	4.67	0.02	0.42	0.29
N use efficiency ^5^, %	30.7	31.7	29.2	27.9	0.51	0.01	0.78	0.06
N use efficiency ^6^, %	29.1	32.0	27.2	21.4	2.22	0.03	0.54	0.09

^a–c^ Means within a row with different superscripts differ (*p* < 0.05). ^1^ N: Nitrogen; NH_3_: Ammonia; DM: Dry matter; MUN: milk urea N; PD: purine derivates; ^2^ SEM: Standard error for interaction; ^3^ D: Diversity; SA: Spatial arrangement; D × SA: Interaction; ^4^ Estimated according to Nousiainen et al. [31]; ^5^ Estimated according to Pacheco et al. [35]; ^6^ Milk N efficiency was calculated as milk N ÷ N intake × 100.

**Table 7 animals-10-01301-t007:** Effect of pasture diversity (diverse or simple) and spatial arrangement (mix or separated) on blood parameters of dairy cows.

Items ^1^	Diverse		Simple		SEM ^2^	*p* Values ^3^		
	Mix	Separated	Mix	Separated		D	SA	D × SA
Hematocrit, %	33.0 ^ab^	34.3 ^a^	34.7 ^a^	31.0 ^b^	1.01	0.36	0.15	0.01
**Metabolism**								
Glucose, mM	3.94	3.83	3.87	3.82	0.07	0.57	0.31	0.74
Cholesterol, mM	5.12	5.32	5.25	4.93	0.28	0.65	0.83	0.38
NEFA, mM	0.29	0.28	0.35	0.28	0.05	0.50	0.40	0.57
**Liver function**								
Albumins, g/L	36.8	37.9	38.5	37.2	0.50	0.32	0.84	0.06
Total Bilirubin, µM	1.75	2.07	2.10	1.85	0.14	0.62	0.82	0.08
AST/GOT, U/L	113.0	121.6	123.2	117.9	4.74	0.51	0.74	0.19
Haptoglobin, g/L	0.20	0.18	0.24	0.18	0.01	0.11	0.05	0.14
**N metabolism and kidney**								
Creatinine, µM	66.4	67.2	68.9	67.9	1.11	0.20	0.94	0.45
Urea, mM	3.46 ^b^	2.48 ^b^	3.49 ^b^	5.22 ^a^	0.36	0.01	0.33	0.01

^a,b^ Means within a row with different superscripts differ (*p* ≤ 0.05). ^1^ AST/GOT: Aspartate aminotransferase/glutamate oxaloacetate transaminase; NEFA: Non esterified fatty acids; ^2^ SEM: Standard error for interaction; ^3^ D: Diversity; SA: Spatial arrangement; D × SA: Interaction.

**Table 8 animals-10-01301-t008:** Effect of pasture type on methane emissions and their relationship to animal productivity.

Items ^1^	Mixed Diverse	Mixed Simple	Separated Simple	SEM ^2^	*p* Values ^3^
CH_4_, g/d	335	393	382	20.4	0.22
CH_4_, g/kg of DMI	18.8 ^b^	23.3 ^a^	21.2 ^a^	0.88	0.05
CH_4_, g kg of milk	16.2	15.4	16.2	1.33	0.90
CH_4_, g/kg of FCM	13.9	13.7	13.3	1.22	0.94
CH_4_, g/kg of milk protein	403	409	378	25.3	0.82
CH_4_, g/kg of milk fat	318	318	298	29.7	0.86

^a,b^ Means within a row with different superscripts differ (*p* ≤ 0.05). ^1^ CH_4_: methane; DMI: dry matter intake; FCM: 4% fat-corrected milk; ^2^ SEM: Standard error for interaction; ^3^ D: Diversity; SA: Spatial arrangement; D × SA: Interaction.

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
