# Peer review of "Milk Production, N Partitioning, and Methane Emissions in Dairy Cows Grazing Mixed or Spatially Separated Simple and Diverse Pastures"

_animals, 2020, doi:10.3390/ani10081301_

Round 1

Reviewer 1 Report

The authors have provided some rebuttal, but have not made changes to the paper that would help a reader answer the concerns that were provided.

For example, the Authors state in their rebuttal to the first point:

“More importantly, neither pre-grazing nor post grazing pasture masses in different treatments were too high or low to confound the DM intake of cows. This also can be seen in grazing hours in foraging behavior section. It is hard to imagine that the biological importance of 300 kg above 1400 kg DM ha was a major issue.”

I agree with this – and it is exactly my point.  The authors state that there was no difference based on a p value, without regard to the biological importance of the result.  Why did the authors not add this to the manuscript?

I do not agree with the authors assertion that simply because p>0.05, it is appropriate to state that there was “no difference” or that something “did not affect” something else.    The Mix Cows had a difference of 1.2kg DMI intake per day – whilst not statistically significant, this might be biologically important as it could relate to a liter of milk.

I repeat my comment: The authors use P>0.05 as justification for claiming that there was “no difference” in a number of places through the text, which I will not list. They need to state a biologically important difference and state the power of the experiment to find that difference before making such statements.

The conclusions continue to overstate the data.  "The current study confirmed that increasing pasture diversity and spatial separation of pasture species in adjacent plots increase the proportion of desirable pasture species" occurred in this case, with this particular setup.  It is simply not true to imply that it would always be the case.

It shows a disappointing engagement with the peer review process that the power analysis requested by both reviewers was provided in rebuttal, but not added to the manuscript.

As almost none of the previous comments have resulted in meaningful changes to the text, I recommend against publication.

Author Response

We highly appreciate the service and valuable comments of the reviewers. Regarding the comments of the referee 1, we understand the frustration but we also needed more tangible suggestions to address the issues in our revised version

Reviewer 2 Report

Review of Manuscript Animals-833761

The study aimed at evaluating the effect of different grazing strategies (mixed or spatially separated) on milk production, N efficiency and methane emission in dairy cows. The paper is very interesting and relevant for sustainable dairy milk production nowadays. Moreover, is based on multiple parameters and methods and complete measurements. However, I would recommend the authors to consider the following remarks and I ask them to address each of my comments in their response:

Major comments

In general, the abstract is very limited and delivers no much information, is not clear and do not reflects the outcome of the study. I would recommend to describe better the treatments and provide the results giving some numbers to have an idea of the magnitude of the outcomes.

Introduction was well written and authors could justify the realization of the experiment.

Authors described M&M with sufficient detail, well structured, is easy to follow and I am confident that experiment and measurement were well performed. However, it is not really clear the allocation of cows to the treatments (see L 116-121). If each group of the 12 group of cows were allocated to one plot (12 plots) during the 21-d grazing period. I wonder when the cows were allocated together to the bigger plots of simple mixed and diverse mixed. Were they allocated in another grazing period? How were allocated the plots in terms of forage composition? By looking at the figure 1, it seems that one plot was composed of only one species. Or did one plot of 0.6 ha include all species e.g. in the diverse separated. This issue must be clearly explained to avoid confusions. Figure 1 is confusing and must be improved. One weakness of this experiment is the analyses of N in faeces after oven drying which could have promoted volatilization of N as ammonia.

Results are well described. I am just concerned about interpretation of the P-values around 0.05. According to authors, significance was pretended to be declared at P<0.05, but authors write about differences by P=0.05, which is wrong. This would change some interpretations of results and authors have to amend them.

In the discussion, the results were properly and scientifically highlighted and discussed. Authors could scientifically discuss their findings or provide readers with interesting interpretations of results or giving hypothesis for possible explanations. Some weaknesses and limitations related to the experiment and methods used were mentioned and briefly discussed or justified.

Not so relevant, but I read so many references. I think this can be reduced to the most important. Authors should check whether all references are really necessary

Minor comments

L18: What do you mean with simple? Is confusing. Maybe use the term “pure” or something similar. Diverse is also too general and not clear. Or try to define shortly this terms in the introduction for readers that do not perfectly know this terms.

L19: what kind of environmental impact? Specify

L30: Why simple mixed? If mixed then is not anymore simple. Confusing. Check the terms. Moreover, give more information/description about treatments, it is not really clear in the abstract

L31-34: Too much qualitative information. I would like to read come numbers to have an idea of the magnitude of the results. You have some space in the abstract. What secondary metabolites were measured? Specify

L84: You aimed also at evaluating the effect on feed intake, according to the abstract. Metabolism of what? Too general

L103-107: what were approximately the proportions for the mixed pastures?

L114: Is this a figure or a table?

L122: How was this estimated? Explain

L211: For N content determination in faeces, analyses should be normally run in fresh faeces or after freeze-drying to avoid volatilization of ammonia as drying in an oven. I think, the procedure used here could have affected the contents of N in faeces

L316: 3.3% or percent points? Check

L317: Instead of giving P>0.05, give always the P value of the table. In this case P=0.05. Check also in other instances

L334: P < or = to 0.05? Please check. In table I read P=0.05, and according to what you stated in L283, statistical significances are declared at P<0.05. Therefore, this is contradictory and authors have to check because this change the interpretation in many parameters. See also L317, 345, 354,361

L357: What you have as FCM/DMI is better defined in the literature as feed efficiency. I would recommend to use this term, and the FCM/DMI as explanation in the footnote

L371: The N outputs calculations according to Nousiainen and Pacheco have to be briefly explained in the M&M. N efficiency as well. I wonder also, how urine volume was estimated. Using creatinine? I think this was not explained and is relevant to prove the reliability of N partitioning calculations

L387 and 394: Like the previous observation if P = 0.05, then it was “not greater”. Be careful with this kind of interpretations

L423: Here, results were not statistically analysed through a 2x2 factorial. This must be highlighted in M&M

L431: Was here the P-value for CH4 (g/kg of DMI) exactly or below 0.05. Based on the table, p-value was 0.05, therefore, superscripts must be deleted

L439: I think this is too much speculative, and cannot be concluded based on the results observed here. Be careful with such a conclusions. I think this is more the effect of the chemical composition of forages like you stated e.g. L446 or better environment tin the rumen for the microbes. The latter is reflected in a tendency in milk protein

L460: Do they only “bind” protein or probably reduce the activity of microorganisms, and therefore, reduces the protein degradation?

L622: Probably, benefits can be seeing in long term experiments or also on productivity or sustainability of pasture on long terms. This factors can be also included

Author Response

Please see attached the letter for details responses.

Reviewer 3 Report

General comments:
The manuscript covers an interesting approach to improved pasture management using a suitable experimental design (as far as I can judge, see issues below). It is well written and mostly easy to read. However, description of experimental design and statistics can be improved and several small spelling/grammar issues should be addressed. Materials and Methods as well as Results are described in a comprehensive manner. Discussion is ample and addresses all findings and original questions. I like the idea of chosing conclusion-like sentences as subheadings in the discussion section. However, in cases where conclusions cannot be derived unequivocably, there is a risk that these subheadings imply findings that are not (clearly) supported by actual results (see e.g. my comments on energy utilization). Please check carefully.
General remark: at some points, use of terminology must be improved. Please check consistency in the chosen wording and terms. See particular comments on terms describing area/pasture types etc. below and please also check terms used for plant secondary metabolites. Currently, several terms are used which are not clearly defined by the authors and as far as I understand, all mean more or less the same regarding the current experiment (natural phytocompounds/bioactive compounds/secondary metabolites/secondary compounds).
General question: Are NDF/ADF data expressed exclusive of residual ash?
Currently, there is no consistent use of superscripts to depict differences among treatment means in tables!
Title: I think „Metabolic Status“ is not covered well by the research methods and in the discussion. In addition, the term is not even mentioned in the whole manuscript. Maybe using „selected blood parameters“ or similar wording in the title would give a better impression of what has been done.
Reference section: Consistency of lower/upper case for first letters in article titles? Journals in abbreviated form or full name? Publisher is not always given when citing book chapters. Please check all references. MDPI refer to ACS style guide in their author instructions.
Below, I have listed more detailed line-specific comments on the manuscript. Besides a number of minor issues mostly regarding language, please take into account my comments on statistics, terminology, discussion on energy utilization and correct citation!
Regarding language, please particularly check section „4.2.1. Shift of N in milk in diverse pasture is only partly driven by level of CT“ for language (grammar and style).
Some sections in my manuscript version were highlighted in yellow, which I treated as the rest.
Line-specific comments:
L 17: „natural phytocompounds“ sounds very unspecific and at the same time complicated. What would this mean?
L 26: In my opinion, „superior“ is too strong, could be replaced by „improved“
L 32: If word limit for the abstract permits, secondary metabolites could be replaced by condensed tannins and total phenolic compounds as these two groups are the specific compounds analyzed.
L 36: Again, „superior“ is too strong. See comments on secondary metabolites/compounds before.
L 64/65: Readability: Can this sentence be rewritten so that the longish part „due to its diuretic effect leading to diluted urinary N concentration“ is not interrupting the main sentence?
Line 65-67: I cannot find any of those statements in O’Connell and Fox (2001) or Cheng et al. (2017). In fact, Cheng et al. (2017) even found reduced urination frequency for plantain compared to other treatments. Of course I did not and cannot check all references but a critical reader might feel encouraged to do so and at this point, I strongly recommend to check all references in the manuscript if they contain the information as suggested.
L 84: The present study aimed to vs. It is hypothesized (L 86) > I would suggest chosing one tense here.
L 87: „with highly bioactive compounds“ or „with high concentrations of bioactive compounds“? Apart from this, I suggest being more specific here (as mentioned before), directly referring to CT and TP.
L 98: „Each group of cows“? (given n=3, and also consistent with terminology used later (e.g. L 116))
L 102: A 7.2 ha was used to conduct... Something is missing here (plot, area?).
L 101: Experimental design and grazing management: Description of plot arrangement and experiment could be improved. Maybe separating the preparation and planning of pasture (including sizing and design of single plots) and actual experiment (cows per block) could be make the description easier to understand.
L 114: Figure 1: I recommend giving all three replicated blocks in a figure, as it does not take too much space but will improve understanding of the experiment.
L 116: Each group of 3 cows (2 multiparous and 1 primiparous cows) was randomly assigned...
L 132: DM intake: Difficult to understand how PM was actually used. ...100 measurements in each daily allocation.../20 rising plate meter readings were taken... How do these measurements fit/interrelate? When were calibration quadrats taken? Before animals were allowed to graze the respective area or during the experiment? i.e., did calibration procedure include different stages of growth/height and thus DM, yielding variable data to allow for regression?
L 135: Abbreviation (PM) is already given, so it can be „total of 20 PM readings...“
L 157: Can you very briefly mention the type of method in the text, e.g. „using a [...] method as previously described“
L 160: Reference [23] is specifically about tannins, is this the correct citation for TP analysis?? Given the details that are provided e.g. for analysis of urinary compounds, I suggest extending the description of TP and CT analysis, including a more specific definition of what is and what is not covered with these terms in the current manuscript.
L 163 Can this one-day observations be viewed as representative for the whole 7-d sampling period?
L 179: BCS has not been introduced as abbreviation before, so it should be given as full term here.
L 190 and others: is full stop after subheadings intended?
L 194: Feces were collected via manual stimulation or as they defecated... Not totallly clear: Feces samples were collected as cows defecated upon manual stimulation or by manual (grab) sampling?
L 201: Both Blood analyses and Fecal and urine N analysis are 2.5.2., so numbering should be adjusted.
L 211: „to determine DM content“ could be omitted here because drying was done not only to determine DM content but also as preparation for subsequent analyses, where DM analysis is mentioned again (next sentence).
L 214: was determined using
L 224: ...into 1 mL transparent HPLC vials...
L 236: Quantitation of urea was determined: Sounds like duplicate meaning. Could it be either „Quantitation was achieved/done“ or „Amount ws determined“?
L 251: Did you also calculate milk N use efficiency in the more straightforward way milk N/N intake?
L 259: before placing them in the cows
L 278: Like mentioned before, using group and herd interchangeably may be misleading to the reader. If I am not mistaken, the sentence in L 277 is saying the same as the one starting from L 278 (mean of group of 3 cows is used for ANOVA)?
L 281: How does this analysis (3 levels of pasture treatment as one fixed effect) fit the experimental design and the 2 × 2 factorial model mentioned in L 271 (main effects of diversity and spatial separation)? I thought pasture type (simple vs. diverse) and spatial arrangement (mixed vs. separate) were the main fixed effects?
L 286: 3.1. DMI and Grazing Behavior: Use of capital letters in (sub)headings is not consistent in the current manuscript.
L 307: In the table heading it says „mix swards“ whereas in the text the term used is „mixed pastures“. Similarly, the term „spatially separated strips“ is used here („spatially separated pastures“ elsewhere). I strongly recommend to check terminology for consistency throughout the whole manuscript, as the design is not easy to understand and confusion is added when terms like pasture/sward/area/strip/plot/subplot/allocation/seeding arrangement/sowing arrangement are not used consisently.
L 315: There was no difference...
L 326: How has ME been estimated? Please add in Materials and Methods section.
L 369: consumed 122 g/d more N compared
L 372: in Table 6, superscripts (presumably to differentiate between statistically different results — it is not explained, also in other tables) are given for urea, but nowhere else (e.g. for N). Is this intended?
L 373: Explanation for PD is missing.
L 386: greater than of those that grazed the mixed
L 389: PD was adjusted by creatinine, no differences were observed between pastures
L 394: than those grazing mixed / than those on mixed
L 403: I suggest using either „N use efficiency“ or „N utilization“
L 416: when cows were grazing on diverse
L 424: I suggest using either „methane“ or „CH4“ and then stick to it, currently it is mixed in the manuscript.
L 429: Either „than those grazing mixed“ or „than those that grazed mixed“ > please check the manuscript, this wording occurs several times
L 449: omit „detected“
L 504: energy level
L 526: Is something missing here? (urine concentration of ? decreased >30%)
L 544: In our experiment diverse pastures had only a numerical larger NFC compared to simple pastures. > Sentence sounds odd.
L 551: ...various pastures. The urine...
L 560: provides
L 562: „better microbial rumen activity“ is quite vague. How would you define better in this case? Both higher and lower microbial activity in the rumen can be beneficial, depending on specific questions (e.g. overall feed degradability vs. excessive feed protein degradation).
L 570/571: protein (singular, twice). In L 571, what is rumen protein(s)? Do you mean feed protein in the rumen?
L 583 onwards: Utilization of energy is no simple variable (and has not been measured in this experiment), so although some results point towards an improved utilization of energy in diverse pastures, the conclusion given as subheading 4.3. (L 583) cannot be drawn from the results that clearly.
Has BCS been considered in blocking the cows for the experiment? If not, the conclusion drawn from comparison of BCS may be invalid. Maybe changes in BCS would be more suitable?
Did you estimate total intake of ME (ME concentration multiplied with DMI)? Effects on BCS may also be simply due to energy intake and not (solely) related to efficiency.
However, your observation of lower CH4 emission in cows grazed on diverse pastures support your findings (simply speaking, less methane means more ME). You could link these two aspects in the discussion.
L 619: DMI already exists as abbr.
L 627: To my knowledge, the term „environmental health“ more often refers to human health seen in relation to its environment (public health) than to „health of the environment“ as you probably meant.
L 843: In Proceedings of Proceedings of > probably one „Proceedings of“ is surplus

Author Response

Please find attached the file for our point-by-point responses. 

This manuscript is a resubmission of an earlier submission. The following is a list of the peer review reports and author responses from that submission.

Round 1

Reviewer 1 Report

This experiment is meant to investigate important questions related to energy corrected milk production and environmental impacts comparing access to diverse and simple pasture species and differing pasture management strategies. 

The number of animals per group (3) and the diversity of age within each group (2 mature cows and 1 lactation 1 cow) is concerning.  Animal availability is understandably limiting given the herd size, but confounds the utility of the information.

The article as written is difficult to read and would benefit from better organization of the findings.  Better separation of the findings would improve the reader's experience and the chance of reaching the target audience.  Creating two companion publication would be very useful in this case, but separation of the current submission into Part A and Part B might be equally useful. 

The discussion is extensive, but somewhat speculative in some areas.  Editing and condensing this section would be useful.

Reviewer 2 Report

This paper involved an experiment where 36 cows (blocked for age, liveweight and days in milk, but seemingly not production) were divided into groups of 3 and exposed to one of four different pasture arrangements.

The pasture arrangements consisted of simple (2 species) and diverse (6 species, including the two in the simple group) which were either grown in a mix, or planted as monocultures in the same yard.

I have some reservations about the experimental design and interpretation that preclude my recommending publication.

Just looking at the results section:

3.1. DMI and Grazing Behavior

286: The authors state that there was no difference between pre- and post- grazing pasture masses (P>0.05).  A p value describes the probability that an observed difference was due to chance.  If it's greater than 0.05, we cannot be confident that the difference was not due to chance.  This does not mean there was no difference (ie that they were similar.  If you wish to claim that two parameters are not different, you need non-inferiority or equivalence calculations, or at a bare minimum to define a biologically important difference and to show a power calculation to justify your sample size.  In this case, the differences observed were quite substantial – and may have been biologically important.  It may be that sample size was the reason that there was no statistical difference.

The authors use P>0.05 as justification for claiming that there was “no difference” in a number of places through the text, which I will not list.  They need to state a biologically important difference and state the power of the experiment to find that difference before making such statements.

3.2. Pasture Nutritive Value and Botanical Composition

Same comment as above.  It isn’t right to state that “The diversity of pastures did not affect any nutritive value parameters except CP content of pastures”.  If there were a million replicates, the same differences would be statistically significant.  The issue is with the precision of measurement.

The botanical composition of the mixed and separated strips was different – but I am not sure this is unexpected.  Plants grow at different rates, and compete with each other with varying success. 

The plantain changed from 20.7% to 11.0% and the authors are maintaining that there was no difference merely because of a p value, with no regard to sample size or variation.

Given they were being fed different diets, differences in other parameters are not surprising.

3.3. Milk Production and Composition

The authors maintain that milk production was not affected (for similar p value reasons).  Without some data showing that the milk data were similar at the beginning of the experiment, I don’t see that any valid interpretation can be made of the milk data presented.

3.6 Methane production

It is not surprising that methane production might vary with different diets.  It is unclear from the data [resented whether it was the separation or the diet that affected the methane production.  Why did the authors not do a linear regression to account for differences in (for example) protein % of diet?

The authors conclude  that

  • The spatial separation increased legume 34 and CP content in simple pasture but decreased NDF in both diverse and simple pastures
  • increasing pasture diversity can help improving the animal production while 36 decreasing the environmental effect of dairy farming,

I don’t think these conclusions are valid, or that there is evidence presented they would be applicable in other situations.

The paper has flaws in it’s design, statistical interpretation and conclusions.  I cannot recommend publication.